# Major, Trace and Rare Earth Element Distribution in Water, Suspended Particulate Matter and Stream Sediments of the Ob River Mouth

**Andrei Soromotin** [1,*], **Dmitriy Moskovchenko** [2,3], **Vitaliy Khoroshavin** [3], **Nikolay Prikhodko** [1], **Alexander Puzanov** [4], **Vladimir Kirillov** [4], **Mikhail Koveshnikov** [4], **Eugenia Krylova** [4], **Aleksander Krasnenko** [5] and **Aleksander Pechkin** [5]

1 Research Institute of Ecology and Natural Resource Management, Tyumen State University, 625003 Tyumen, Russia
2 Tyumen Scientific Centre, Siberian Branch of Russian Academy of Sciences, 625026 Tyumen, Russia
3 Institute of Earth Sciences, Tyumen State University, 625003 Tyumen, Russia
4 Institute for Water and Environmental Problems of Siberian Branch of the Russian Academy of Sciences, 656038 Barnaul, Russia
5 Arctic Research Center of the Yamal-Nenets Autonomous District, 629008 Salekhard, Russia
* Correspondence: asoromotin@mail.ru

**Abstract:** Ongoing climatic changes are influencing the volume and composition of the river waters that enter the Arctic Basin. This hydrochemical study was conducted within the mouth of the Ob River, which is one of the world's largest rivers, providing 15% of the Arctic Ocean's total intake. Concentrations of suspended and dissolved elements were determined using ICP–MS and ICP–AES. As compared to the world average values, the Ob river water had higher concentrations of dissolved P, As, Cu, Zn, Pb and Sb, i.e., the elements that form soluble organo-mineral complexes. The composition of suspended matter was characterized by low concentrations of most trace elements (Cd, Cr, Co, Cu, Mo, Al, Ni, Pb, V) due to their low contents in peat soils within the river drainage basin. Concentrations of dissolved forms were many times lower than concentrations of suspended forms in Al, Fe, Mn, Zn, Cr, Co, Ti, Sc, and all rare earth elements. Total concentrations of Ni, Cu, Bi, Pb, W in the river water increased by 2.5 to 4.2 times during the summer. The effects of climate change, which can cause an increase in the discharge of solid particles from thawing permafrost, are likely to lead to an increase in the discharge of certain elements into the Ob River estuary.

**Keywords:** Arctic basin; stream runoff; trace elements; total suspended matter; dissolved elements; bottom sediments

## 1. Introduction

The formation of the composition of the waters of the Arctic Ocean depends to a large extent on the runoff of the largest rivers (the Ob, Lena, and Yenisei). The boreal regions of Russia play a crucial role in the transfer of elements from the continents to the ocean at high latitudes [1,2]. The inflow to the Arctic Ocean of chemical elements in the form of dissolved compounds, suspensions, and bedload sediments is by an order of magnitude more intense compared to other oceans: in spite of the fact that the Arctic Ocean contains only a little more than 1% of the global total oceanic water mass, this ocean receives as much as 10% of the global riverine runoff [3].

Over the past few decades, the climate in Western Siberia has undergone significant changes. An increase in air temperature [4], an increase in the depth of seasonal ground thawing [5,6], and an increase in the thickness of snow cover [7] have been revealed. In most of the Ob basin, except for the steppe zone, there is an increase in precipitation [8]. Climate warming is the main reason for the increase in runoff of the rivers of the Arctic basin, which also has also increased the transport of chemical elements. Climate change affects the chemical

composition of river water directly through changes in the temperature regime and indirectly through changes in the hydrological cycle and transformation of geochemical processes in the watershed [9]. This increase in seasonal thawing changes the composition of soils by increasing the migration of organic carbon and trace elements [10–12]. As a consequence, the composition of river water changes. Compared to the years 1981–1990, the runoff of organic matter with the Ob water increased 1.3–1.5-fold in 1991–2000 [13]. Warming, causing an increase in the depth of seasonal thawing, increases the proportion of the suspended form of elements in the runoff and can lead to an increase in the concentration of alkalis and alkaline-earth elements [14]. Measurements of solute concentrations from previously unstudied watersheds throughout west Siberia suggest that warming and permafrost degradation will likely amplify the transport of dissolved solids to the Kara Sea and adjacent Arctic Ocean [15].

At the present stage, there is a statistically significant increase in autumn and winter low-water runoff in the lower Ob [16]. In the 21st century, warming in the Arctic is likely to be stronger than in any other ecosystem on earth, which will have serious consequences for terrestrial and aquatic biogeochemical cycles [17]. By the middle of the 21st century, river flow in the Arctic Basin is projected to increase by 4–14% [18]. Therefore, one of the most important problems of biogeochemistry is to obtain information about changes in the flux of macro- and microelements in river runoff from the continent to the ocean, particularly in the Arctic region, where the most significant effects of climate change are expected [19].

Recently, data characterizing the content of dissolved and suspended forms of chemical elements in the mouth and the estuary of the Ob have been obtained [20–24]. The components analyzed included mainly nutrients determining the biological productivity of water areas and heavy metals as the most common inorganic pollutants. In this connection, it is of undoubted interest to expand the range of studied elements in the composition of the runoff of Arctic rivers [3]. Data on the content of trace elements in large river basins are necessary to assess the impact of climate change and land use; however, sufficiently large-scale studies of such a plan are rare [25].

During 2020–2021, we conducted hydrochemical studies in the low course and the mouth of the Ob River. The main aim of the study was to determine the Ob river input of chemical elements to the Arctic basin. This investigation has the following objectives: (i) determine the content of macro-, trace-, and rare-earth elements in the water and carry out calculations of elemental discharge into the Ob estuary; (ii) determine the ratio of suspended and dissolved forms of elements; (iii) evaluate the features of element deposition by comparing the suspended and bottom sediment composition. Since modern climate changes have affected the hydrological regime and changed the runoff during the summer–autumn and winter low-water periods [16], the studies were carried out from these two hydrological seasons.

## 2. Materials and Methods

### 2.1. Study Area

The Ob is one of the largest rivers on Earth, flowing through the West Siberian Plain. The catchment area (2,570,000 km$^2$) is the sixth largest in the world and the basin is more than 2400 km long in the meridional direction and 2200 km in the latitudinal direction within Russia [8]. The average runoff volume is 402 km$^3$ a year, or about 15% of the total freshwater flow into the Arctic Ocean [26]. The climate is continental, with average temperatures of $-30$ to $-16$ °C in January and $+4$ in July (Arctic zone) to $+20$–$22$ °C (steppe and forest-steppe). The Ob catchment is composed mainly of Quaternary sedimentary formations [27]. In the north, tundra vegetation grows, while in the south, steppes and forest-steppes, for the most part ploughed up, are widespread. Wetlands form a significant part of the watersheds in the middle and lower reaches. The West Siberia Lowland contains one of the largest peat resources on the planet, constituting 12.9% of the global peatlands [28,29]. Waters on peatland are characterized by very low salinity, acidic reaction, and high organic matter content [30]. More than half of the territory of the West Siberian Plain belongs to the permafrost zone,

the southern boundary of which is located between 61–62° N [8]. The land use structure of the watershed is classified as follows: arable lands (36%), forests (30%), wetlands (11%), pastures (10%), shrublands (5%), cultivated (5%), and irrigated arable lands (3%) [31].

The population in the Ob basin is about 27 million people [32]. Numerous oil fields are concentrated in the Ob basin, which together provide 2/3 of Russia's oil production [31]. The development of oil fields has led to significant pollution of the river and changes in the ecosystem [33]. Currently, water consumption in the Russian part of the Ob basin is about 9.5 km$^3$ per year, and 7.7 km$^3$ is discharged as sewage water back into the rivers [34].

## 2.2. Sample Collection

The sampling site is located 8 km upstream of Salemal settlement at the entrance to the river delta, where the Ob channel has minimal branching and the flow is concentrated in one channel stream, which makes this site most representative for studying water flow and substance transfer. The line gauge is located in the narrowing of the channel (Figure 1), and its length is 1325 m. The right bank of river is steep; the excess of the terrace edge over the water edge is up to 3–5 m; the left bank is flat with an excess of 0.4–0.8 m over the water edge at the time of research. Depth surveys with Garmin GPSmap 420 s chart plotter and hand-held chart plotter showed that near-shore sections are 5–8 m deep on average; 50 m from the right bank and 150 m from the left bank there are steep slopes to the main channel; the central flat part of the channel is 24–26 m deep. The channel cross-section area is 26,120 m$^2$. The bottom is composed of pebbles and boulders (right bank), and the left bank is composed of dense silt-clay sediments.

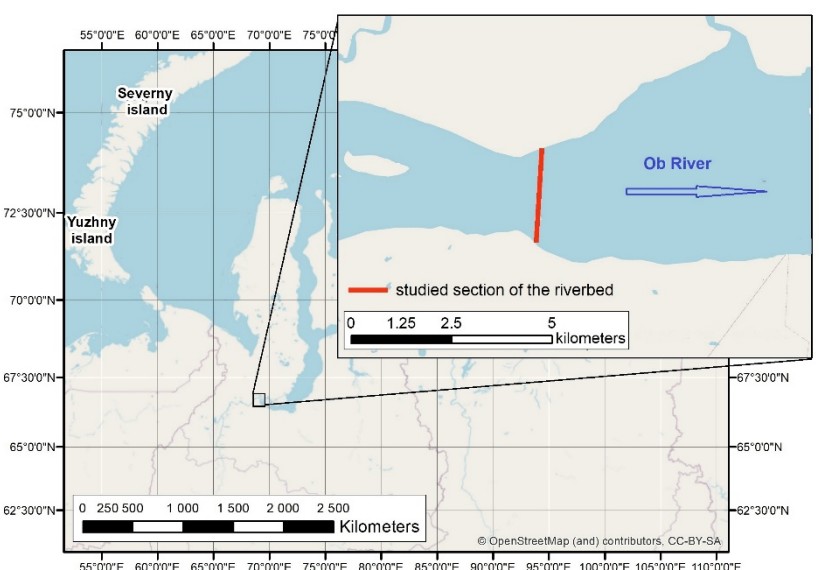

**Figure 1.** Overview scheme of the location of the study area in the lower reaches of the Ob River.

Hydrochemical and hydrometric studies were conducted in two stages. The first stage was the study of water composition and flow rate of the Ob during the summer–fall low-flow period (25–29 August 2020). During the second stage, the water composition was studied at the end of the ice period (22 March–4 April 2021). The current rate was measured using a GRS-3 device (NPO Typhoon, Obninsk, Russia). An electromagnetic sensor is used in the device to measure the water flow rate. In previous studies [35], high accuracy of measurements and efficiency of the device application for evaluation of hydrological characteristics of rivers were shown. Flow rate were measured by two- and three-point methods, depending on the depth of the channel at the measuring point. At depths up to 10 m, measurements were made at 0.2 h, and 0.8 h, where h is the depth of the riverbed. At depths more than 10 m, the measurements were made at the marks 0.2 h, 0.5 h, 0.8 h. To assess the accuracy of the measurements, we compared the obtained results with

the information from the Service for Hydrometeorology and Environmental Monitoring (Roshydromet) for Salekhard, formed in the dataset of the Arctic Great Rivers Observatory (ArcticGRO) scientific group [36]. The comparison showed that in the Salekhard site, which is located 140 km upstream of our observation site, the runoff data differ slightly, by 3–7%.

Water sampling was performed using a Ruttner sampler every 100 m along the length of the station. At each station, water was sampled from three depths (0–20 cm, 0.2 h, and 0.8 h, where h is the river depth). In 2020, samples were also taken at intermediate verticals located 50 m from the main from the surface layer (0–20 cm) and in depths of 0.6 h. Samples were poured into plastic bottles and immediately sent to the laboratory. Bottom sediment sampling was conducted using a Petersen dredger in the same bed section.

### 2.3. Sample Analysis

The pH, salinity (TDS), conductivity (EC), and dissolved oxygen (DO) were determined in situ using a WTW 3420 multimeter (Xylem Analitics, Weilheim, Germany).

In the laboratory, part of the samples was filtered through pre-weighed nitrocellulose Millipore™ filters with a diameter of 47 mm and a pore size of 0.45 μm into 15 mL polypropylene tubes. The fraction passing through the filter was considered dissolved. Although the mass of a trace element that passes through a 0.45-μm filter is not necessarily dissolved, as the trace element may be associated with colloidal phases < 0.45 μm [37], we hereby use the term "dissolved" for simplicity of discussion. A measure of 0.2 mL of ultrapure concentrated nitric acid ($HNO_3$ 65% Suprapur, Merck) was added to each sample. The unfiltered samples were also poured into polypropylene tubes with the addition of ultrapure nitric acid.

The chemical composition of water samples was investigated by the methods of inductive coupled plasma mass spectrometry (ICP-MS) and inductive coupled plasma atom emission spectrometry (ICP-AES). The contents of the macrocomponents (Na, Mg, Al, P K, Ca, Fe) and some trace elements (Li, V, Cr, Ti, Mn, Co, Ni, Cu, Zn, Sr, Ba) were determined by ICP-AES (iCAP 6500, Thermo Scientific, Waltham, MA, USA). Contents of trace elements and rare earth elements (REEs) were determined by ICP-MC Thermo Elemental-X7 spectrometer (Thermo Scientific, Waltham, MA, USA). Several elements (Li, Ti, V, Cr, Co, Mn, Cu, Zn, Sr, and Ba) were determined by two methods to check the correctness of the analysis. In all cases, the discrepancy in the content of these elements determined by two methods did not exceed 15%. For elements which were determined both by ICP-AES and ICP-MS, average values were used. In addition, to check the correctness of the analysis of samples, there was used a Certified Reference Material (CRM) "Trace Metals in Drinking Water" produced by High-Puriy Standards (USA). The discrepancy with the CRM did not exceed 6%. The methods, detection limits, and analytical results of the CRM in water samples are given in the Supplementary Materials (Table S1).

Filters containing insoluble sludge were weighed on laboratory analytical scales after drying in a desiccator at t = 80 °C to determine the amount of total suspended matter (TSM) in river water. In the suspended matter and in bottom sediments, the content of chemical elements was also determined by ICP-AS and ICP-MC methods. The analyzed samples were prepared by acid digestion in an open beaker system. The samples (20–50 mg of insoluble sludge and 100 mg of the bottom sediments) were placed in Teflon beakers (volume 50 mL) together with 0.5 mL $HClO_4$ (Perchloric acid fuming 70% Supratur, Merck), 3 mL (HF Hydrofluoric acid 40% GR, ISO, Merck), and 0.5 mL of $HNO_3$ (Nitric acid 65%, max. 0.0000005%% GR, ISO, Merck) and 0.1 mL of solution containing 8 μg dm$^{-3}$ 145 Nd, 161 Dy, and 174 Yb isotopes which were necessary to control completeness of digestion. The samples with acids were boiled and evaporated until intense white vapors appeared. The beakers were cooled, their walls were washed with water, and the solution was again evaporated to wet salts. Then, 2 mL of HCl (Hydrochloric acid fuming 37% OR, ISO, Merck) and 0.2 mL of 0.1 M $H_3BO_3$ solution were added and evaporated to a volume of 0.5–0.7 mL. All chemicals were analytical grade reagents. The resulting solutions were transferred into

polyethylene bottles, 0.1 mL of a solution containing 10 mg L$^{-1}$ In (internal standard) was added, diluted with deionized water to 20 mL, and analysis was performed.

To check the accuracy of measurements, we used multi-element certified reference materials Trapp ST-2a (Russian State Standard GSO 8671-2005). The comparison with the standard samples showed a sufficient repeatability (85–115%) for the majority of the analyzed elements, except for Sn, Mo, Ba, Ag, and W, the measurements of which were excluded from the calculations. The methods, detection limits, recoveries, and analytical results of the CRM are given in the Supplementary Materials (Table S2).

### 2.4. Statistical Treatment

Statistical processing of the results included calculation of averages, standard deviation (SD). The differences in element content between different seasons was assessed using the Mann–Whitney test ($p < 0.05$). To assess geochemical features of suspended solids and bottom sediments, *EF* enrichment coefficient values were calculated by the formula:

$$EF = \frac{Cx}{CAl}(sample) / \frac{Cx}{CAl}(crust),$$

where *Cx* (*sample*) is the measured concentration of the element of interest, *Cx* (*crust*) is the concentration of the same element in the Earth's crust, and *CAl* is the concentration of the reference element (aluminum) in the same sample and the Earth's crust. The composition of the upper continental crust was used as a reference for normalization due to the lack of sufficient data on the composition of soils and rocks of the catchment area. All calculations were performed in Excel 2019.

## 3. Results

### 3.1. Flow Rate and Physicochemical Parameters

The average flow rate of the Ob River as of 27 August 2020 was 0.50 m·s$^{-1}$. With a channel cross-section of 26,120 m$^2$, the discharge was about 13,300 m$^3$·s$^{-1}$. On 3 April 2021 at the end of the ice season, the mean Ob channel current speed was 0.18 m·s$^{-1}$, which means a water flow of 4970 m$^3$·s$^{-1}$ with a channel cross-section of 27,608 m$^2$. Water discharge in winter was 2.7 times less than in summer. Flow values, in comparison with the data of the previous years [36], were average, typical for the last decade.

The pH value depends on the content of HCO$^{3-}$ and H$^+$ ions and the concentration of CO$^2$ in the river water. At the end of the ice period, water reaction was neutral, the average pH value = 6.97 (Table 1).

**Table 1.** Physico-chemical parameters of the Ob River water.

| Index | August 2020 (*n* = 60) | April 2021 (*n* = 30) |
|---|---|---|
| pH | 8.01 (7.52–8.87) | 6.97 (6.90–7.00) |
| TDS, mg L$^{-1}$ | 116.0 (50–307) | 190.7 (186–194) |
| TSM, mg L$^{-1}$ | 21 (13–34) | NM |
| Color index | 87.6 (65–93) | 29.9 (28.8–30.7) |
| EC, μS | 151 (143–158) | 251 (242–253) |
| DO mg O$_2$ L$^{-1}$ | 8.7(7.8–9.3) | 6.2 (4.2–9.9) |

Note: Mean and min-max (in parentheses); NM—Not measured.

Due to activation of phytoplankton life activity in the warm period, the average pH value increased to 8.01 units due to decrease of CO$_2$ content going to photosynthesis. In summer, stratification of pH distribution at different depths was observed, from 8.07 in the surface layer to 7.88 in the bottom layers (Figure 2a). The maximum pH values in the surface layer are caused by the greatest development of living organisms in it. The TDS value varies regularly throughout the year. During the ice period, when the river is fed by groundwater, the average TDS value was 190.7 mg L$^{-1}$. In summer, the TDS value

decreases due to atmospheric precipitation to 116 mg $L^{-1}$. The highest TDS value was observed in the near-bottom layer (Figure 2b).

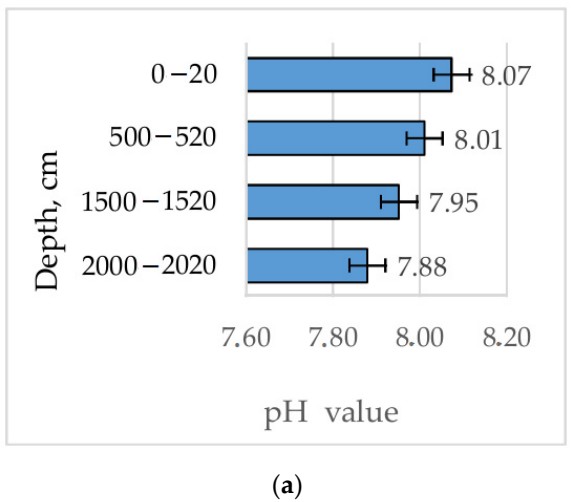 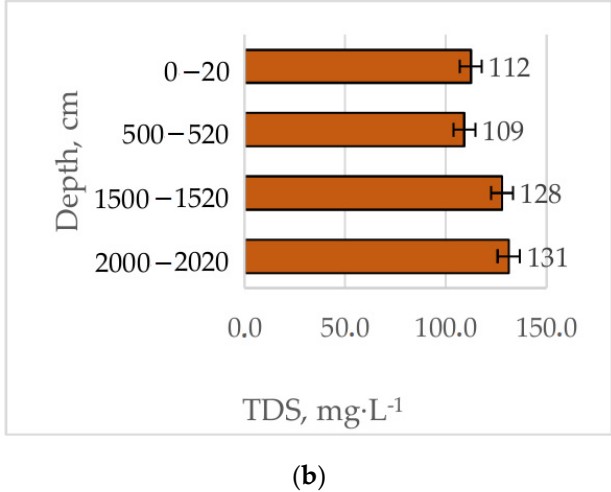

(**a**)                                        (**b**)

**Figure 2.** Distribution of pH and TDS (total dissolved solids) at different depths: (**a**) pH value at different depths; (**b**) TDS value at different depths. Whiskers represents the standard error.

The obtained values of physicochemical parameters are typical for waters of the Ob and its downstream tributaries. The usual values of pH of Ob River waters in the taiga zone are 7.0–8.5 units in summer [38]. The TDS value of waters of the Ob and downstream tributaries is 100–200 mg $L^{-1}$ in the taiga zone and <100 mg $L^{-1}$ in the tundra zone [39].

Water chromaticity in summer was almost 3 times higher than in winter (Table 1). The color of river water is determined by chromophores of total organic carbon and organic iron [40]. The high color values indicate that bogs, which occupy up to 80% of the territory in some areas of the watershed, enrich Ob waters with humic and iron-containing compounds in summer.

*3.2. Elemental Composition*

The total concentration of chemical elements in the unfiltered water samples and elemental flux in the two surveyed hydrological seasons are shown in Table 2. For convenience, the elements were divided into 3 groups: macroelements, trace metals, and metalloids and rare earth elements.

Ca prevails among macroelements in the composition of the Ob river waters, followed by Na, Mg, and Si in descending order of their concentrations. The prevalence of Ca is typical of the Ob river waters almost throughout its course except for the headwaters located in steppe zone [8]. The concentration of Na stays within the average content of 10,000–20,000 µg $L^{-1}$ for this element during low runoff [38].

The contents of some elements surpass the limit values of water quality set in Russia. For instance, the content of Fe is dozens of times higher than the maximum permissible concentration (100 µg $L^{-1}$). In particular, it is notably high in the ice period. Similarly, the maximum permissible concentration of Mn (10 µg $L^{-1}$) is exceeded by orders of magnitude in winter. The increased content of these elements is caused by natural reasons such as their high contents in bog waters. Acid leaching of metals (Fe and Mn) from peatlands is enhanced in surface waters of the West Siberia with waterlogged catchments [9]. The contents of Mn and Fe in this study correspond to the average values for the Ob river. It has been noted that Fe content of 2200 µg $L^{-1}$ and Mn content of 170 µg $L^{-1}$ are typical for the river [38].

**Table 2.** Elemental composition of Ob River water and element flux (unfiltered samples).

| Element | August 2020 (n = 60) | | | March–April 2021 (n = 30) | | |
|---|---|---|---|---|---|---|
| | Mean | SD | Fluxes, t·day$^{-1}$ | Mean | SD | Fluxes, t·day$^{-1}$ |
| Macroelements, µg L$^{-1}$ | | | | | | |
| Ca | 19,141 | 1108 | 21,995 | 35,672 | 295 | 15,318 |
| Fe | 762 | 159 | 876 | 3030 | 431.5 | 1301 |
| K | 920 | 45.4 | 1057 | 1194 | 29.1 | 513 |
| Mg | 4859 | 348 | 5584 | 8386 | 76.2 | 3601 |
| Na | 6831 | 468 | 7850 | 11,921 | 253 | 5119 |
| P | 75 | 12.6 | 86 | 102 | 14.0 | 44 |
| Si | 2811 | 255 | 3230 | 7840 | 849 | 3367 |
| Trace metals and metalloids, µg L$^{-1}$ | | | | | | |
| As | 1.58 | 0.19 | 1.8 | 1.42 | 0.07 | 0.61 |
| Ba | 22.2 | 3.53 | 25.5 | 34.2 | 0.95 | 14.7 |
| Be | 0.0156 | 0.0063 | 0.018 | 0.0017 | 0.0040 | 0.0007 |
| Bi | 0.0054 | 0.0017 | 0.006 | 0.0014 | 0.0003 | 0.0006 |
| Cd | 0.012 | 0.0057 | 0.013 | 0.013 | 0.0049 | 0.0056 |
| Co | 0.17 | 0.07 | 0.20 | 0.85 | 0.06 | 0.36 |
| Cr | <DL | - | - | 0.11 | 0.58 | 0.05 |
| Cu | 1.94 | 0.28 | 2.2 | 0.68 | 0.15 | 0.29 |
| Li | 2.6 | 0.17 | 3.0 | 3.9 | 0.039 | 1.7 |
| Mn | 37.9 | 13.4 | 43.6 | 645 | 54.4 | 277 |
| Mo | 0.37 | 0.028 | 0.43 | 0.39 | 0.019 | 0.17 |
| Ni | 1.62 | 0.24 | 1.9 | 0.39 | 0.31 | 0.17 |
| Rb | 0.77 | 0.049 | 0.88 | 0.82 | 0.02 | 0.35 |
| Pb | 0.39 | 0.09 | 0.45 | 0.15 | 0.06 | 0.06 |
| Sb | 0.094 | 0.027 | 0.11 | 0.061 | 0.018 | 0.03 |
| Sr | 114 | 6.12 | 131 | 206 | 1.83 | 88.5 |
| V | 1.58 | 0.25 | 1.8 | 0.67 | 0.05 | 0.29 |
| Zn | 2.25 | 0.93 | 2.6 | 1.29 | 0.51 | 0.55 |
| Rare earth elements, ng L$^{-1}$ | | | | | | |
| Ce | 644.9 | 299.3 | 0.74 | 263.5 | 21.4 | 0.11 |
| Dy | 65.4 | 15.5 | 0.08 | 28.2 | 2.1 | 0.012 |
| Er | 34.9 | 7.9 | 0.040 | 16.2 | 1.0 | 0.007 |
| Eu | 17.3 | 4.4 | 0.020 | 6.4 | 0.6 | 0.003 |
| Gd | 80.9 | 19.6 | 0.093 | 34.0 | 2.3 | 0.015 |
| Ho | 12.8 | 2.9 | 0.015 | 5.7 | 0.4 | 0.002 |
| La | 346.7 | 155.1 | 0.40 | 118.0 | 9.4 | 0.051 |
| Nd | 368.8 | 93.8 | 0.42 | 136.9 | 9.8 | 0.059 |
| Pr | 85.6 | 21.8 | 0.10 | 30.6 | 2.0 | 0.013 |
| Sm | 77.8 | 19.4 | 0.09 | 30.5 | 1.7 | 0.013 |
| Tb | 11.8 | 2.8 | 0.014 | 4.9 | 0.4 | 0.002 |
| Tm | 4.6 | 1.0 | 0.005 | 2.3 | 0.2 | 0.001 |
| Y | 334.5 | 73.6 | 0.38 | 145.8 | 8.3 | 0.063 |
| Yb | 29.4 | 6.5 | 0.034 | 15.6 | 1.1 | 0.007 |

Note: Contents of Se, Ga, Ge, Rh, Pd, Hg, Ag, Te, Ta Re, Os, Ir, Pt, Au in the studied samples were below their detection limits; n—number of samples; < DL—below detection limit.

The average content of Zn is 1.23 µg L$^{-1}$ in winter and 2.25 µg L$^{-1}$ in summer, which are close to the values observed in the middle Ob (2.5 µg L$^{-1}$) [41]. The average content of Cu in the studied samples was slightly lower than the value of 3 µg L$^{-1}$ observed in the middle Ob [38]. Generally, the composition of the Ob river water in this study was typical for the middle and lower reaches, where the inflow of water from the boggy catchment areas is a key factor.

### 3.3. Fluxes of Elements

Our estimates of average constituent fluxes for the two periods (August 2020–March–April 2021) are presented in Table 2. The calculations determined the seven major elements

(Ca, Na, Mg, Si, K, Fe, and Mn) contribute 99% of the total mass load of elements to the Ob estuary. Ca, Mg, Si, and Na had the largest loads, exceeding 3 thousand tons per day.

To assess the reliability of the obtained values of the fluxes, we conducted a comparison with the data given in the generalized study on the hydrochemistry of Ob [22] (we recalculated the values from ton-year$^{-1}$ to ton-day$^{-1}$). The values of Ca and Mg flux obtained by us in summer period practically coincide with the previous data (Figure 3).

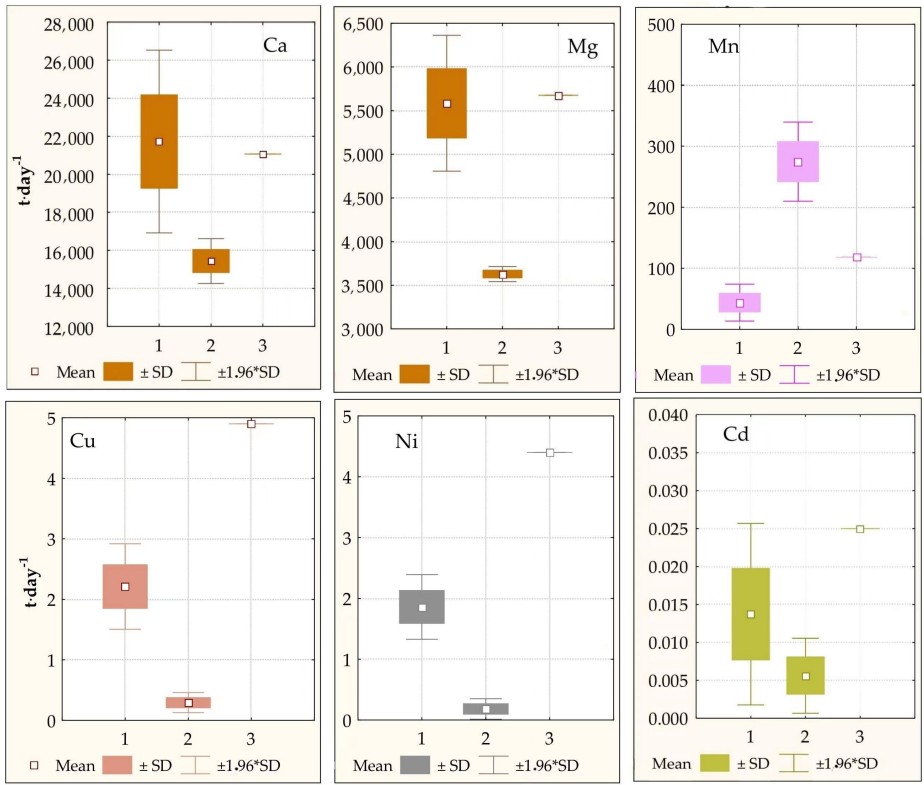

**Figure 3.** Total flux (t·day$^{-1}$), mean value of Ca, Mg and trace elements with Ob waters: 1—summer–fall low-water period 2020 (our data); 2—ice period 2021 (our data); 3—average for Ob River (data from [22] was used).

Discharge in winter is significantly less than in summer (our measured values were 4970 and 13,300 m$^3$/s, respectively), so the flux of almost all chemical elements into Ob mouth in summer is greater than in winter. Especially significant are the differences for Cu, Ni, Bi, Pb, V, which flux increases 5 to 10 times in summer (Table 2). The exception is Fe, Mn, and Co, the flux of which is greater in the ice period than in summer, since high concentrations of these metals in this period compensate the decrease in water flow volumes with an excess.

Manganese flux, according to our data, has greater than average values in winter and lower in summer. The content of heavy metals (Cu, Ni, Cd) is 2–2.5 times less in summer and 4.5–25 times less in the ice period. This is probably due to the relatively small amount of suspended solids during our studies, the average content of which was 21 mg L$^{-1}$, which is approximately two times less than the average value of 40 mg L$^{-1}$ for Ob [16]. The maximum amount of suspended matter arrives during the flooding period, which was not covered by our studies. Probably, therefore, the values of Cu, Ni, and Cd runoff, according to our data, turned out to be low. Thus, many chemical parameters showed distinct seasonal patterns.

*3.4. Seasonal Dynamics*

There was significant difference (at $p < 0.05$) in element concentrations between seasons. In summer, the concentration of Cu, Pb, Bi, Zn, Sb, Ni, and all REEs increases. In winter,

the concentration of macroelements, Fe, Mn, Co, Cr, Li, Ba, Sr, is higher than in summer. The only exceptions are As, Be, Cd, Mo, and Rb, which exhibited approximately the same values in summer compared to the ice season.

The reasons for the seasonal dynamics of the elemental composition of the Ob and tributary waters are well enough investigated. The fact of summer consumption of Si in dissolved form by diatomic algae is widely known [42]. Therefore, the decrease in the concentration of Si during the summer period should be associated with the biogenic factor. It is a well-known fact that mineralization and the content of major ions ($HCO_3^-$, $SO_4^2$, $Cl^-$, $Ca^{2+}$, $Mg^{2+}$, $Na^+$, $K^+$) increase during the ice period [8,22].

The mineralization of Ob water and the content of substances in dissolved form are inversely related to the magnitude of water discharge: maximum mineralization is usually observed at the end of the ice period, when rivers are fed by groundwater [38]. The increased content of dissolved substances in groundwater runoff compared to surface runoff is associated with a more intense ion exchange: in the northern regions of Western Siberia, groundwater interacts with mineral rocks, and surface water with a thin layer of summer thawed peat containing little soluble salts. An increase in water salinity leads to an increase in the content of dissolved forms of metals, which is caused by the formation of strong complex compounds with the mineral component of water. Thus, when studying the effect of salinity on the content and distribution of Cd, Cu, Pb, Zn in the Ob River, it was found that with increasing salinity, due to the formation of strong chloride complexes that hold the trace metals in the water column, the content of dissolved forms of trace metals increases and suspended decreases [43].

It was noted that the content of Fe, Mn in the dissolved form in winter in Ob waters is higher than in summer [22]. Seasonal differences in the content of these metals are largely related to redox conditions. The formation of reducing conditions in bottom sediments at the end of the ice period leads to the reduction of $Mn^{4+}$ to $Mn^{2+}$ and its intense influx into the water [44]. In summer, the decrease in Fe and Mn concentration is associated with the deposition of amorphous hydroxides to the bottom with increasing pH values, as well as active absorption from the solution and suspended matter by biosorbent organisms [23].

However, the described reasons for seasonal variability concern the dissolved form of elements, which is characteristic only for waters with a small amount of suspended solids (<5–10 mg $L^{-1}$ [45]). In our studies, higher values of TSM were obtained (13–34 mg $L^{-1}$, see Table 1), so the seasonal dynamics are also associated with differences in the amount of suspended solids carried by the river. It is known that suspended solids content in Ob water is minimum in winter when the average TSM = 10 mg $L^{-1}$, whereas in the warm period, it increases up to 50 mg $L^{-1}$ [16]. It can be assumed that the summer increase in the content of many elements (Cu, Ni, Pb, V, Zn, et al.) is associated with an increase in the runoff of insoluble particles. The seasonal dynamics of the content of elements in the dissolved form are less significant in quantitative terms than the growth of the flow of suspended matter. Obviously, the seasonal variability of solid runoff is strongest for low-soluble, hard-hydrolysable elements that migrate predominantly in the suspended form. For example, 80% of Pb migrates in suspended form during the autumn season in western Siberian rivers [14].

Thus, the seasonal dynamics of the total content of chemical elements are determined by a complex of factors. The observed multiple increases in the content of Mn, Fe, and Co, close in geochemical properties, during the ice period are associated with the seasonal dynamics of acid-base and redox conditions. The content of easily hydrolysable macroelements (Na, Mg, Ca, K) depends on the inflow of groundwater in winter. The content of Si, Ca, Fe is influenced by increased activity of aquatic biota in summer. Summer growth of Cu, Ni, Pb, V, Zn concentration depends on an increase of suspended solids inflow.

### 3.5. Partitioning of Elements

The content of metals in dissolved and suspended forms is presented in Table 3. In early studies, it was noted that Fe, V, Mn, and Ni in Ob are mainly contained in suspended

matter, while Cu and Zn are distributed in approximately equal proportions between suspended and dissolved forms [46]. It was later clarified that 50 to 97% of Mn, Zn, and Pb in Ob waters are in the insoluble particles, and Cu, Cd, and As are predominantly in the dissolved state [22,23]. Recent studies of watersheds in the north of Western Siberia have shown that in small and medium-sized rivers there can be distinguished 3 categories of elements: (i) soluble, highly mobile elements, <20% of which migrate as particles and >80% in the dissolved state (Li, B, Si, Na, Mg, K, Ca, Ni, As, Rb, Sr, Mo, Sb, and Ba); (ii) insoluble elements, whose insoluble forms are >60% (Al, Ti, Fe, Ga, Zr, Nb, Cs, REEs, Hf, Pb, Th); and (iii) elements of medium solubility whose insoluble forms are 20–60% (V, Cr, Mn, Co, Cu, Zn, Cd, Y, U) [14]. The dissolution and existence of these metals in ionic form are promoted by low pH values and the increased content of humus substances, which is characteristic of bog waters [45].

**Table 3.** Content of chemical elements in suspended and dissolved forms, Ob, August 2020 (n = 9).

| Element | TSM | | TDM | | WA RDM | AEA RDM | Ratio TSM/ (TSM + TDM) |
|---|---|---|---|---|---|---|---|
| | Mean | SD | Mean | SD | | | |
| Macroelements, µg L$^{-1}$ | | | | | | | |
| Al | 586 | 375 | 13.3 | 8.9 | 32 | 78 | 98 |
| Ca | 3020 | 1741 | 15,332 | 2553 | 14,600 | - | 16 |
| Fe | 1119 | 380 | 16.0 | 12.3 | 66 | 271 | 99 |
| K | 115 | 73 | 930 | 37 | 1350 | - | 11 |
| Mg | 152 | 78 | 4887 | 91 | 3800 | - | 3 |
| Na | 79.0 | 38.9 | 7096 | 203 | 5100 | - | 1 |
| P | 51.1 | 11.1 | 22.8 | 9.2 | 9 | 12 | 69 |
| Si | 299 | 168 | 2510 | 85 | 4070 | 2540 | 11 |
| Ti | 31.6 | 18.8 | <DL | - | 0.489 | 0.0012 | ND |
| Trace metals and metalloids, µg L$^{-1}$ | | | | | | | |
| As | 0.41 | 0.12 | 1.1 | 0.06 | 0.62 | 0.77 | 27 |
| Ba | 9.2 | 3.89 | 13.4 | 2.3 | 23 | 14.4 | 41 |
| Co | 0.20 | 0.11 | <DL | - | 0.148 | 0.073 | ND |
| Cr | 1.21 | 0.44 | <DL | - | 0.7 | 0.47 | ND |
| Cu | 0.67 | 0.14 | 1.8 | 0.21 | 1.48 | 1.29 | 27 |
| Ga | 0.11 | 0.07 | <DL | - | 0.03 | 0.025 | ND |
| Li | 0.30 | 0.18 | 2.69 | 0.11 | 1.84 | 2.92 | 10 |
| Mn | 32.6 | 15.3 | 0.5 | 0.38 | 34 | 18.7 | 98 |
| Nb | 0.09 | 0.06 | <DL | - | 0.017 | - | ND |
| Ni | 0.66 | 0.29 | 1.3 | 0.14 | 0.801 | 0.89 | 34 |
| Pb | 0.63 | 0.25 | 0.9 | 1.1 | 0.079 | 0.118 | 34 |
| Rb | 0.67 | 0.42 | 0.75 | 0.19 | 1.63 | 1.14 | 47 |
| Sr | 13.7 | 7.6 | 101.3 | 10.7 | 60 | 149 | 12 |
| V | 0.39 | 0.14 | 0.8 | 0.1 | 0.71 | 0.82 | 33 |
| Zn | 3.05 | 1.25 | <DL | - | 0.6 | 4.2 | ND |
| Zr | 0.86 | 0.50 | 0.06 | 0.02 | 0.039 | 0.176 | 93 |
| ng L$^{-1}$ | | | | | | | |
| Cd | 16.5 | 7.0 | 19.0 | 11.3 | 80 | 15 | 46 |
| Hg | 0.9 | 0.3 | <DL | - | - | - | ND |
| Mo | 11.0 | 2.4 | 394 | 18 | 420 | 270 | 3 |
| Sb | 45.0 | 34.0 | 193 | 53 | 70 | 72 | 19 |
| Th | 88.3 | 46.8 | 1.8 | 0.3 | 41 | 26 | 98 |
| U | 25.9 | 10.0 | 277 | 33 | 372 | 145 | 9 |
| W | 16.3 | 7.7 | 59 | 147 | 100 | 12 | 22 |
| Rare earth elements, ng L$^{-1}$ | | | | | | | |
| Ce | 803.5 | 368.5 | 13.3 | 8.6 | 262 | 328 | 98 |
| Dy | 69.1 | 26.5 | 2.4 | 1.1 | 30 | 33 | 97 |
| Er | 38.7 | 14.5 | 2.3 | 1.0 | 20 | 19 | 94 |
| Gd | 84.4 | 32.6 | 1.9 | 0.8 | 40 | 44 | 98 |
| La | 417.7 | 173.8 | 8.3 | 5.4 | 120 | 177 | 98 |
| Nd | 410.0 | 166.0 | 6.7 | 4.6 | 152 | 213 | 98 |

**Table 3.** *Cont.*

| Element | TSM | | TDM | | WA RDM | AEA RDM | Ratio TSM/ (TSM + TDM) |
|---------|------|-----|------|-----|---------|---------|------------------------|
| | Mean | SD | Mean | SD | | | |
| Pr | 97.8 | 40.4 | 1.8 | 0.9 | 28 | 50 | 98 |
| Sc | 120 | 70 | <DL | - | 1200 | - | ND |
| Tb | 11.9 | 4.5 | 1.1 | 0.4 | 5.5 | 6 | 92 |
| Ho | 13.5 | 5.2 | 0.9 | 0.3 | 7 | 6.3 | 94 |
| Sm | 89.1 | 35.6 | 1.8 | 0.8 | 36 | 44 | 98 |
| Yb | 34.0 | 13.8 | 1.8 | 0.8 | 17 | 12 | 95 |

Note: WA RDM—World average river dissolved matter [47]; AEA RDM—Arctic regions of Eurasia average river dissolved matter [24]; < DL—below detection limit; ND—no data; n—number of samples.

According to our data, the content of Al, Fe, Mn, Zn, Zr, Sc, Th, and all REEs in suspended form in the Ob mouth is many times, by 1–2 mathematical orders, higher than in the dissolved state. This group of elements also includes Cr, Co, Ti, Ga, Zn, Sn, whose content in dissolved form was below the detection limit. The content of Cd and Rb is distributed approximately in equal proportions; the dissolved form of Ni, Pb prevails insignificantly. The other elements migrate mainly in the dissolved form. Thus, the Ob mouth differs from small rivers flowing through wetland watersheds in low content of dissolved zinc, which is probably due to higher pH values of water.

To assess the regional hydrochemical features, we compared our data with the data of previous studies on the average composition of the Ob [22], the rivers of Eurasia's Arctic basin [22,24], and the world average data [47]. Our data on dissolved forms of macroelements (Ca, Mg, K, Na) in the Ob mouth are close to the world average values (Table 3). The content of Si is approximately twice less, and the amount of phosphorus, on the contrary, is 2.5 times greater. The decrease in the content of soluble form Si in the surface layer of marine waters and waters of the Ob estuary is associated with its uptake by phytoplankton and diatom algae [21,41]. Probably, a similar process operates in the waters of the lower Ob. A sharp decrease in the Si content in the Ob estuary due to the reproduction of diatom algae was also noted earlier [48]. The increased content of mineral phosphorus is associated with its high solubility in an acidic environment, which is characteristic of the soils of the northern regions of Western Siberia. Generally, in an acidic environment the main P bearing compounds are remarkably soluble [49]. A high As content was noted in the water of the Irtysh, the main tributary of the Ob, and was explained by the distribution of rocks containing large amounts of phosphates at the river head [50].

The content of soluble As, Cu, Pb, and Sr, i.e., chalcophile elements capable of forming soluble organomineral complexes, was higher than the world average [51]. The increased concentrations are due to their intensive leaching from acid soils of the northern regions of Siberia. In addition to the acidic mobilization of metals, dissolved organic matter from soils may act as ligand and keep metals in solution. Significant discrepancies between the average zinc concentrations in the Ob ($3.05 \ \mu g \ L^{-1}$) and of the world ($0.6 \ \mu g \ L^{-1}$) waters are most likely due to a mistake in determining the latter value, since previous estimates gave concentrations of $30 \ \mu g \ L^{-1}$ [52], and the dissolved copper content in river waters only in rare cases exceeds the zinc concentration [24].

Content of the dissolved form of Mn, Fe, Al, Th, Tl, and REEs is well below the world average (Table 3) and average values for Arctic rivers (Figure 4). The content of many elements (Co, Cr, Zn, Ga, Hg, etc.) was below the detection limit, which also indicates low concentrations. In the summer period, when sampling was carried out, the content of dissolved forms of most heavy metals in waters of Ob is less than in winter [22]. The content of dissolved forms of iron and manganese was particularly low in our results. It is known that the ratio of dissolved and suspended forms of iron and manganese elements is controlled by redox conditions. Under reducing conditions, Fe and Mn are in the soluble form, while under oxidizing conditions, they pass into insoluble precipitate. Therefore, the

suspended form of these elements absolutely prevails in the samples analyzed by us in the summer period, and the content of the dissolved form is reduced. Absorption of dissolved heavy metals by phytoplankton, which is most active in the summer period, also reduces their levels in water bodies [14,53].

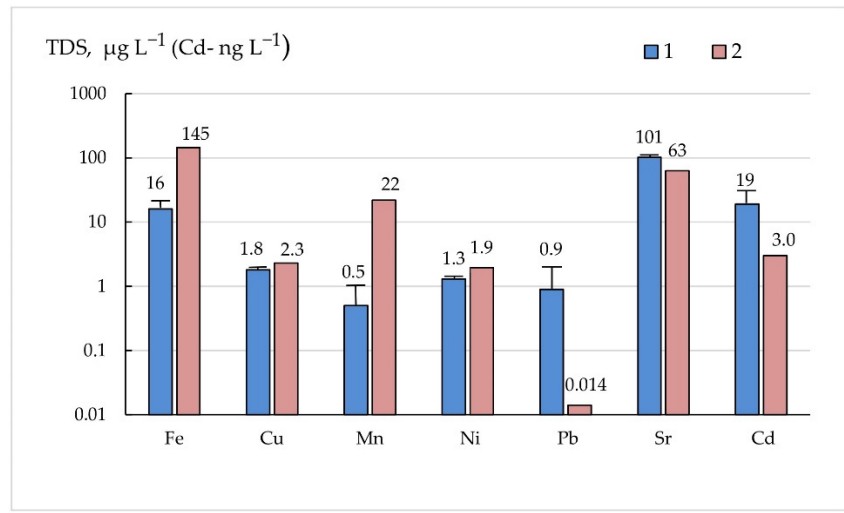

**Figure 4.** Content of the soluble form of elements in Ob waters: (logarithmic values): 1—the year 2020 autumnal low-water period (our data); 2—Ob River average (data from [22] were used). Error bars represent 1 SD.

The content of trace elements (As, Cd, Cu, Fe, Mn, Pb, Zn) that migrate as part of the insoluble suspension, according to our data, was 1.5–2.5 times less than in other rivers of the Arctic basin (Figure 5).

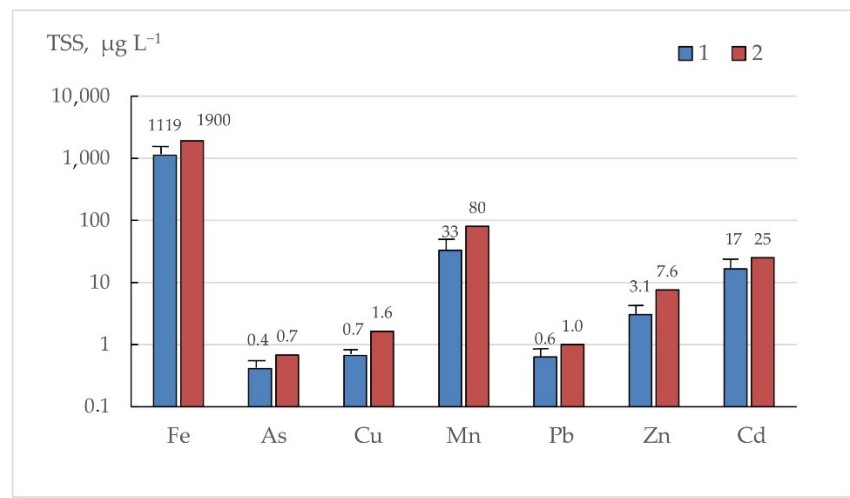

**Figure 5.** Content of insoluble form of elements in river waters (logarithmic values): 1—Ob River, summer–autumn low water 2020 (our data); 2—Arctic rivers of Eurasia average (data from [22] were used). Error bars represent 1 SD.

Probably, the reduced value is connected with the hydrological season: in summer–autumn low water, the suspended solids content in the river water is several times less than in the period of high water. Thus, according to our data, the average quantity of TSM was 21 mg L$^{-1}$ during the period of studies, while in the lower reaches of the Ob river, the average value of suspended solids content is 52 mg L$^{-1}$ [16]. Thus, a 2.6-fold decrease in suspended sediment flux compared to average values resulted in a 1.5–2.5-fold

decrease in total trace element content. Suspended forms of metals expressed in volumetric values ($\mu g\ L^{-1}$ or $ng\ L^{-1}$) are distributed in the lower Ob in proportion to the content of the suspension itself. A similar conclusion was made based on the distribution of metals in the suspension in the Ob estuary at the exit to the Kara Sea [23]. It should also be noted that the Ob is a plain river draining areas of flat taiga and tundra relief with permafrost and low thickness of the weathering crust, which leads to a lower content of the suspended sediment itself [22,54].

Climate change scenarios in the polar and circumpolar regions of Western Siberia suggest that permafrost thawing will intensify. One of the consequences will be an increase in the flow of suspended solids into the waters of rivers and lakes [14]. Active removal of silt particles is confirmed by an increase in the turbidity of tundra river water, which is associated with thermokarst, thermoerosion, and slope processes [55]. Thus, we should expect an increase in the intake of elements in insoluble form, particularly Hg and Cd, the content of which is increased in the surface horizons of soils in the north of Western Siberia [56].

### 3.6. Influence of Suspended Matter on Bottom Sediment Composition

An analysis of the chemistry of bottom sediments reveals geochemical properties of watershed as well as actual long-term state of water bodies contamination. In contrast to volumetric concentrations, weight concentrations of elements (% or $\mu g\ g^{-1}$) provide an indication of the geochemical properties of the suspension itself and determine its influence on the composition of bottom sediments.

The weight concentration of macroelements in the river sediment of Ob decreases in the series Ca > Fe > Al > Mg > K > Na > P > S > Mn ~ Ti (Table 4). The exceptionally high content of Ca (average 13.6%) indicates the predominant influence of biogenic material. The maximum biomass in the Ob's water in the lower reaches is observed at the end of summer, when a huge amount of zooplankton is carried into the main river channel from the lowlands, so-called "sors", flooded during the period of flooding [57]. Thus, the detected high concentration of biogenic Ca in suspended solids has a seasonal character.

**Table 4.** Elemental composition of suspended solids and bottom sediments, Ob River ($n = 10$).

| Elements | Suspended Solids | | | | | Bottom Sediments | | |
|---|---|---|---|---|---|---|---|---|
| | Mean | SD | EF | WARSM | AEA RSM | Mean | SD | EF |
| | | | | Macroelements, % | | | | |
| Al | 2.70 | 0.95 | 1.0 | 8.7 | 5.9 | 6.09 | 1.03 | 1.0 |
| Ca | 13.6 | 6.0 | 16.1 | 2.59 | 1.28 | 0.97 | 0.11 | 0.5 |
| Fe | 5.62 | 1.92 | 4.3 | 5.8 | 8.0 | 3.18 | 0.44 | 1.1 |
| K | 0.52 | 0.17 | 0.7 | 1.69 | 1.34 | 1.74 | 0.05 | 1.0 |
| Mg | 0.70 | 0.09 | 1.4 | 1.26 | 0.93 | 0.69 | 0.16 | 0.6 |
| Mn | 0.16 | 0.05 | 6.2 | 0.2 | 0.22 | 0.15 | 0.10 | 2.5 |
| Na | 0.37 | 0.09 | 0.5 | 0.71 | 0.63 | 1.07 | 0.15 | 0.6 |
| P | 0.26 | 0.09 | 12.1 | 0.2 | 0.41 | 0.074 | 0.020 | 1.5 |
| Ti | 0.15 | 0.05 | 1.2 | 0.44 | 0.46 | 0.43 | 0.09 | 1.4 |
| | | | | Trace metals and metalloids, $\mu g\ g^{-1}$ | | | | |
| As | 21.5 | 10.7 | 13.5 | 36.3 | 36 | 6.95 | 4.00 | 1.9 |
| Bi | 0.97 | 1.93 | 18.4 | 0.85 | - | 0.14 | 0.03 | 1.1 |
| Cd | 0.90 | 0.64 | 30.3 | 1.55 | 0.55 | 0.16 | 0.07 | 2.4 |
| Cr | 60.6 | 19.9 | 2.0 | 130 | 108 | 60.5 | 11.0 | 0.9 |
| Co | 9.5 | 3.55 | 1.7 | 22.5 | 24 | 20.4 | 9.47 | 1.6 |
| Cu | 34.9 | 13.3 | 3.8 | 75.9 | 46 | 16.5 | 3.18 | 0.8 |
| Ga | 5.2 | 1.97 | 0.9 | 18.1 | - | 11.9 | 1.67 | 0.9 |
| Hf | 1.16 | 0.35 | 0.66 | 4.04 | - | 2.33 | 0.46 | 0.6 |
| Hg | NM | NM | - | - | 0.05 | 0.07 | 0.01 | 1.8 |
| Li | 13.3 | 3.4 | 1.9 | 8.5 | - | 20.3 | 3.9 | 1.3 |

Table 4. *Cont.*

| Elements | Suspended Solids | | | | | Bottom Sediments | | |
|---|---|---|---|---|---|---|---|---|
| | Mean | SD | EF | WARSM | AEA RSM | Mean | SD | EF |
| Ni | 32.0 | 11.0 | 2.1 | 74.5 | 52 | 39.2 | 8.33 | 1.1 |
| Rb | 30.8 | 10.6 | 1.1 | 78.5 | - | 70.2 | 5.87 | 1.1 |
| Nb | 4.01 | 1.45 | 1.0 | 13.5 | - | 8.55 | 1.25 | 1.0 |
| Pb | 33.9 | 23.2 | 6.0 | 61.1 | 34 | 14.8 | 1.6 | 1.2 |
| Sb | 2.66 | 3.41 | 20.1 | 2.19 | - | 0.63 | 0.04 | 2.1 |
| Sr | 620 | 236 | 5.9 | 187 | 145 | 170 | 13 | 0.7 |
| Ta | 0.39 | 0.39 | 1.3 | 1.27 | - | 0.62 | 0.09 | 0.9 |
| Tl | 0.15 | 0.05 | 0.51 | 0.53 | - | 0.31 | 0.02 | 0.5 |
| Th | 4.20 | 1.37 | 1.21 | 12.1 | - | 6.82 | 1.05 | 0.9 |
| U | 1.30 | 0.55 | 1.5 | 3.3 | - | 1.57 | 0.22 | 0.8 |
| V | 71.4 | 26.0 | 2.2 | 129 | 134 | 89.5 | 13.8 | 1.2 |
| Zn | 156 | 82.3 | 7.0 | 208 | 263 | 59.0 | 11.4 | 1.2 |
| Zr | 40.4 | 14.0 | 0.6 | 160 | 118 | 90.1 | 17.0 | 0.6 |
| Rare earth elements, µg g$^{-1}$ | | | | | | | | |
| Ce | 38.9 | 12.3 | 1.9 | 73.6 | 32 | 60.3 | 6.8 | 1.3 |
| Eu | 1.00 | 0.31 | 3.0 | 1.29 | 0.86 | 1.03 | 0.08 | 1.4 |
| Gd | 4.18 | 1.40 | 3.2 | 5.25 | 3.3 | 4.03 | 0.31 | 1.3 |
| La | 20.5 | 6.6 | 2.0 | 37.4 | 21 | 26.7 | 2.3 | 1.2 |
| Lu | 0.24 | 0.08 | 2.3 | 0.35 | 0.29 | 0.27 | 0.02 | 1.2 |
| Nd | 20.2 | 6.6 | 2.3 | 32.2 | 9.5 | 24.1 | 1.9 | 1.2 |
| Pr | 4.80 | 1.54 | 2.0 | 7.95 | - | 6.21 | 0.53 | 1.2 |
| Sc | 5.45 | 1.86 | 1.2 | 18.2 | - | 10.2 | 1.85 | 1.0 |
| Sm | 4.40 | 1.47 | 2.8 | 6.12 | 1.7 | 4.82 | 0.36 | 1.4 |
| Y | 19.2 | 6.12 | 2.8 | 21.9 | - | 18.3 | 1.40 | 1.2 |

Note: Contents of Se, Rh, Pd, Te, Re, Ir, Pt, Au in the studied samples were below their detection limits; WARSM—World average river suspended matter [58]; AEA RSM—Arctic regions of Eurasia average river suspended matter [22]; NM—not measured.

Calculation of the EF coefficient showed that the suspension is strongly enriched in Ca, Cd, P, As, Bi, and Sb (EF = 12.1–30.3). Similarly, in the main tributary of the Ob river, the Irtysh, significant enrichment of suspended solids of Cd and As was revealed (Gordeev et al., 2004). Suspensions of Fe, Mn, Pb, and Zn were slightly less enriched (EF = 4.3–7). Practically, for all elements values EF > 1, that is connected, first of all, with small sizes of particles. The grain size distribution of SPM gradually decreases from the finest particles (<4 µm) to the most coarse ones (>50 µm) [22]. Metals are mainly accumulated in fine-grained sediments [59]. The main reason is that the smaller particles have a higher surface-to-volume ratio [60].

The Fe/Al ratio in the river sediment indicates the influence of the geological and landscape structure of the watershed. Thus, the Fe/Al = 0.61 ratio is typical of silicate rocks, while Fe/Al = 2 indicates the influence of peatlands [50]. According to our data, the Fe/Al ratio = 2.1 for suspended matter that indicates the influence of peatlands. In bottom sediments, Fe content is approximately 2 times lover than Al content, typical for the silicate rocks. Probably, the increased content of some trace elements in the suspension is connected with the ingress of particles of bog genesis. The peat of West Siberian bogs contains an increased content of Cd [56] and As [61], which correlates well with the increased content of these elements in the river suspension. Organic iron dissolved and colloidal compounds flocculate during bog water transport and adsorb dissolved P and As [50].

The content of most trace elements and REEs in the Ob sediment is low compared to the world average. Extremely low concentrations are recorded for Mo, Al, Zr, Ga, Nb, Hf, Ta, Th, and Tl. The ratio of content in Ob to world average values for most heavy metals (Cd, Cr, Co, Cu, Ni, Pb, V ets) varies within 0.3–0.6 (Figure 6). Concentrations of Fe, Mn, and Y are close to the world average values. Only Ca, Sr, P, Li, Sb, and Bi accumulate relative to the world average values. Similar results (stronger depletion for Zr,

Mo; impoverishing for Al, V, Cr, Ni, Cu, Ga, Zn and high concentrations of Mn and P) were obtained when analyzing the suspended sediment composition of small and medium-sized rivers in the West Siberian Lowland [14].

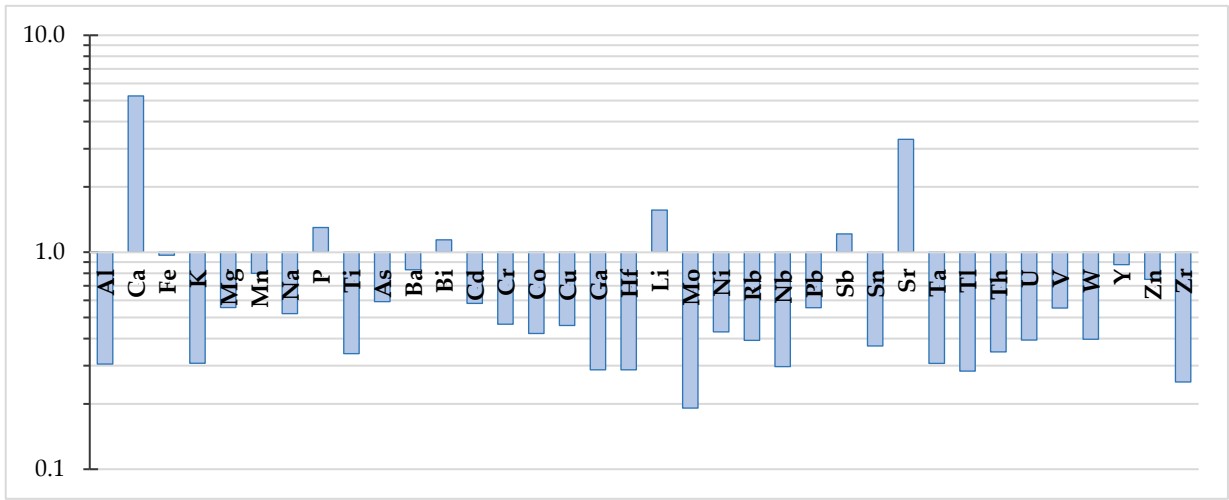

**Figure 6.** Ratio of concentration of elements in the sediment of Ob (our data) to the average concentration of elements in the sediment of the rivers of the Earth (for calculations, we used the data from [58]).

The low content in the Ob River sediment corresponds to the low content of elements in the soils of the catchment area, especially in the northern part. The soils of the north of Western Siberia, as reported in [56], are characterized by low concentrations of most heavy metals, which are 3–9 times lower than the world average. The following factors also play a role [14]: (i) Low runoff and low suspended sediment concentrations because peatlands have no rocks and mineral substrate subject to physical weathering; (ii) the organic rather than mineral nature of the surrounding "solid" substrates and the consequent organic rather than silicate nature of the suspension, and (iii) the high concentration of dissolved organic matter and Fe, leading to high concentrations of typically low-soluble elements in the dissolved (<0.45 μm) fraction due to colloids.

The average EF values of Ti, Ba, Cu, Zn, Cr, Mo, Ni, Li, Pb Mo, and all REEs in bottom sediments were <1.5, indicating that these metals come from natural weathering products [62,63]. Higher EF values were noted for Mn (2.5), Cd (2.4), Sb (2.1), As (1.9), Hg (1.8). EF values in the interval of 2–5 belong to the category of moderate enrichment and moderate pollution [64]. When comparing the composition of the suspension and bottom sediments (see Table 4), it is obvious that the concentration of Cd, Sb, As is significantly higher in the suspension. The high content of Cd and As was noted earlier in the sediment of the Irtysh River, the largest tributary of the Ob [50]. It was concluded that the input of As was caused by natural processes, i.e., weathering of phosphate-bearing sediments with high concentrations of As in the upper Irtysh and high affinity As-P in boggy peat soils. Dissolved P and As are absorbed by the amorphous organic components of Fe oxyhydroxide, which act as carriers during transport to the main channel of the Irtysh River. In contrast, Cd enrichment is mainly due to anthropogenic influences. The threefold enrichment of Cd suspended solids near the large industrial city of Omsk (Ob basin) and the subsequent decrease to background levels at 500–700 km downstream quite definitely points to municipal and industrial wastewater as the main source of Cd [ibid]. However, we cannot unequivocally assert an anthropogenic source of Cd in the sediment samples studied by us in the Ob mouth, because this element is intensively accumulated by sphagnum mosses, the main peat formers of Western Siberian bogs [65]. This accumulation is caused by high adsorption capacities of divalent metals on Sphagnum [66]. This enrichment

exhibits a universal character since it is detected at the northern and southern sites of Western Siberia [29].

## 4. Conclusions

Study of water chemistry in the Ob River showed significant differences in the chemical composition of the Ob River in summer and ice periods. Seasonal dynamics of hydrochemical indicators consists in an increase of total content of macroelements (Ca, Mg, Na, K, Si Fe) and some microelements (Mn, Co) in winter, which is caused by inflow of ground water in winter, decrease of water pH, and formation of reducing conditions in bottom sediments. In summer, a significant role is played by biogenic processes and the growth of suspended solids. Discharge is minimal during ice season, which leads to a decrease of flux of almost all elements in this hydrological season in comparison with summer–autumn low-flow period, except for Fe and Mn, because high concentrations of these metals in winter compensate for a decrease in volume of water flow. In summer, the flux of Cu, Ni, Bi, Pb, V, W, Zr, and all REEs into the Ob River mouth increases 5–11 times compared to the ice period; Cd, Mo, and Sr increases 1.5–2.5 times.

Studies have confirmed that Al, Fe, Mn, Zn, Zr, Sc, Th, and all REEs enter the Ob estuary mainly in suspended form. Modern climatic changes, which lead to an increase in the flow of insoluble particles into the rivers due to thawing of permafrost, may lead to an increase in the flow of these elements as well as Hg and Cd, both of which are elevated in the widely distributed peat soils of the catchment area. To more accurately predict changes, studies should be supplemented by studies of the flow of insoluble particles during the spring flood, when the flow of suspended solids is maximal.

The water of the Ob River, in comparison with other rivers of the Earth, contains an increased amount of dissolved elements, which in acidic peat soils form soluble organo-mineral complexes. These include As, Cu, Pb, Sb. A high concentration of calcium was noted in the composition of the suspended matter, which indicates a large amount of organic material. Calculation of the EF coefficient showed the enrichment of the suspension with elements that accumulate in the peat (As) or coming from anthropogenic sources (Cd). Judging by the Al/Fe ratio, the main process determining the composition of bottom sediments is silicate weathering, while the suspended matter composition is influenced by the removal from wetland, peat-covered landscapes.

**Supplementary Materials:** The following supporting information can be downloaded at: https://www.mdpi.com/article/10.3390/w14152442/s1, Table S1: Methods of analysis, detection limits, analytical results, and recovery of certified reference material "Trace Metals in Drinking Water" (High-Purity Standards, USA); Table S2: Methods of analysis, detection limits, analytical results, and recovery of certified reference material "Trapp ST-2a (Russian standard GSO 8671-2005)".

**Author Contributions:** A.S.: Methodology, Writing—review & editing, Review and editing and funding acquisition; D.M.: Formal analysis writing, Writing—original draft, Manuscript draft, Data curation; V.K. (Vitaliy Khoroshavin): Conceptualization, Data curation, Investigation; N.P.: Visualization; A.P. (Alexander Puzanov): Supervision, Data curation; V.K. (Vladimir Kirillov): Validation; M.K.: Data curation, Investigation; E.K.: Data curation, Investigation; A.K.: Data curation, Investiga-tion; A.P. (Aleksander Pechkin): Data curation, Investigation. All authors have read and agreed to the published version of the manuscript.

**Funding:** This research was funded by the Yamalo-Nenets Autonomous District Government (West-Siberian Interregional Science and Education Center's project 3II/00069-22/5). This study was also supported by SB RAS, Tyumen Scientific Centre (project no. 121041600045-8).

**Institutional Review Board Statement:** Not applicable.

**Informed Consent Statement:** Not applicable.

**Data Availability Statement:** The datasets generated during and/or analyzed during the current study are available from the corresponding author on reasonable request.

**Acknowledgments:** This work was supported by the Yamalo-Nenets Autonomous District Government. The role of sponsors is to fund expeditions to the Ob River, as well as to fund chemical analyses of the bottom sediment and water samples collected. The authors are especially grateful to Vasiliy Karandashev for the element determination.

**Conflicts of Interest:** The authors declare no conflict of interest.

## Abbreviations

AEA RDM—Arctic regions of Eurasia average river dissolved matter; AEA RSM—Arctic regions of Eurasia average river suspended matter; DL—detection limit; EF = enrichment factor; REE—rare earth elements; SD—standard deviation; TDM—total suspended matter; TSM—total suspended matter; WA RDM—World average river dissolved matter; WA RSM—World average river suspended matter.

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
