# Peer review of "Major, Trace and Rare Earth Element Distribution in Water, Suspended Particulate Matter and Stream Sediments of the Ob River Mouth"

_water, doi:10.3390/w14152442_

Round 1
Reviewer 1 Report
The paper titled "Major, trace and rare earth element distribution in water, suspended particulate matter and stream sediment of the Ob river mouth" is interesting. The experimental design is sound.
However, the paper needs major revision before being suitable for publication.
At present, the structure of the article does not represent the purposes of the study. It is poorly understandable and accessible to the reviewer and the reader.
This is the reason why the article needs a major revision, not in terms of experimental design (further analysis, blanks, etc.) but in terms of arranging what is written so that it relates to the goals of the study.
Lines 80-87 the goals of the study are clear. The section Results and Discussion should be organized according these purposes.
I suggest the authors should split section 2.2. in two distinct sections: 2.2 sample collection, 2.3 sample analysis.
Line 170 rare eeart elements (REEs), please correct the acronym
Lines 167-181 the paragraph should be rewritten in a clear way
Lines 185-197 the acid digestion in an open beaker system can be questionable, due to the lack of control of some crucial variables, e.g., temperature. What is the amount of bottom sediments digested per aliquot?
In results and discussion, the tables can be moved to Supplementary information and graphs can be added to better comprehend the trend and the concentrations of elements analyzed.
Please add the error bars in the graphs or write the error of measurements in the text
Table S1 please check the results of Recovery %. The recovery is the ratio of the concentration of analyte found to that stated to be present. For instance, the Recovery % for Na should be (6000/5975)*100= 100,4% and not 99.6% as reported.
Does <DL stand for under limit of detection? Is the concentration analyzed under Limit of quantification or under limit od detection? Please check. Perhaps, the columns certified value and found were exchanged with each other.
Author Response
Response to Reviewer 1 Comments
Dear Rewiewer! Thank you for your detailed review of the manuscript. We have made corrections according to your comments.
All corrections are highlighted in green in the manuscript.
Responding in detail to the comments
Reviewer’s remarks |
Corrections |
At present, the structure of the article does not represent the purposes of the study. It is poorly understandable and accessible to the reviewer and the reader. This is the reason why the article needs a major revision, not in terms of experimental design (further analysis, blanks, etc.) but in terms of arranging what is written so that it relates to the goals of the study Lines 80-87 the goals of the study are clear. The section Results and Discussion should be organized according these purposes. I suggest the authors should split section 2.2. in two distinct sections: 2.2 sample collection, 2.3 sample analysis. |
The objectives of the study were formulated as follows: (i) determine the content of macro-, trace- and rare-earth elements in the water and carry out calculations of elemental discharge into the Ob estuary; (ii) determine the ratio of suspended and dissolved forms of elements; (iii) evaluate the features of element deposition by comparing the suspended and bottom sediment composition.
In accordance to the goals of the study, we have restructured the article as follows
2. Materials and Methods 2.1. Study area 2.2. Sample collection 2.3. Sample analysis 2.4.Statistical treatment 3. Results 3.1. Flow rate and physicochemical parameters 3.2 Elemental composition 3.3. Fluxes of elements 3.4. Seasonal dynamics 3.5. Partitioning of elements 3.6. Influence of suspended matter on bottom sediment composition
|
Line 170 rare eeart elements (REEs), please correct the acronym |
Corrected |
Lines 167-181 the paragraph should be rewritten in a clear way |
The paragraph is rewritten as follows: The chemical composition of water samples was investigated by the methods of inductive coupled plasma mass spectrometry (ICP-MS) and inductive coupled plasma atom emission spectrometry (ICP-AES). The contents of the macrocomponents (Na, Mg, Al, P K, Ca, Fe) and some trace elements (Li, V, Cr, Ti, Mn, Co, Ni, Cu, Zn, Sr, Ba) were determined by ICP-AES (iCAP 6500, Thermo Scientific, USA). Contents of trace elements and rare earth elements (REEs) were determined by ICP-MC (Thermo Ele-mental - X7 spectrometer, USA). Several elements (Li, Ti, V, Cr, Co, Mn, Cu, Zn, Sr, and Ba) were determined by two methods to check the correctness of the analysis. In all cases, the discrepancy in the content of these elements determined by two methods did not exceed 15%. For elements which were determined both by ICP-AES and ICP-MS, average values were used. Also to check the correctness of the analysis of samples used a Certified Reference Material (CRM) "Trace Metals in Drinking Water" produced by High-Puriy Standards (USA). The discrepancy with the CRM did not exceed 6%. The methods, detection limits and analytical results of the CRM in water samples are given in the Supplementary Materials (Table SM1). |
Lines 185-197 the acid digestion in an open beaker system can be questionable, due to the lack of control of some crucial variables, e.g., temperature. What is the amount of bottom sediments digested per aliquot? |
The weight of the sample is indicated: The samples (20-50 mg of insoluble sludge and 100 mg of the bottom sediments) were placed in Teflon beakers (volume 50 ml) |
In results and discussion, the tables can be moved to Supplementary information and graphs can be added to better comprehend the trend and the concentrations of elements analyzed. |
We have not been able to achieve a satisfactory presentation of the results in the form of graphs. We have obtained results of changes in a large number of elements, which are difficult to fit into a single figure. Especially considering the large variation in values. The application of logarithm leads to the fact that in the graphs the content of elements in different seasons is practically the same. We therefore decided that a presentation in the form of tables would be more appropriate
|
Please add the error bars in the graphs or write the error of measurements in the text |
The variation values are shown in the graphs , e.g |
Table S1 please check the results of Recovery %. The recovery is the ratio of the concentration of analyte found to that stated to be present. For instance, the Recovery % for Na should be (6000/5975)*100= 100,4% and not 99.6% as reported. |
Corrected |
Does <DL stand for under limit of detection? Is the concentration analyzed under Limit of quantification or under limit od detection? Please check. Perhaps, the columns certified value and found were exchanged with each other. |
<DL mean Under limit of detection |
Reviewer 2 Report
In the presented work, the authors investigated concentrations of major, trace and rare earth elements within the mouth of the Ob River under different hydrological and temporal conditions, and studied the effects of wide variety of physicochemical, geo- and biogenic processes on elements concentrations. By analyzing the distribution of elements in water, suspended particulate matter and stream sediments, the authors could also evaluate the ways in which elements are transported to the Ob estuary and their potential origin.
This study deals with an important topic that can be of interest to the journal readers, and the presented results and related discussion are scientifically sound. I therefore recommend accepting the presented work for publication in the Water journal after minor revision, addressing the comments listed below:
Materials and methods:
- Page 4: lines 167-175: There is some discrepancy in the reporting of certain elements that were determined by the two methods (i.e., ICP-OES or ICP-MS). Namely, the list of elements that were determined either by ICP-OES or ICP-MS method (reported in lines 167-173) do not completely match with the elements determined by both methods (listed in lines 174-175).
- ICP-MS/AES instrumental parameters that were used for determination of element concentrations in river samples are missing and should be added to e.g. supplementary material.
Results and discussion:
- Page 6, lines 252-255: Have the authors considered that ICP-MS/AES analysis of river water samples that were just acidified might not reflect the total concentration of elements in the samples? It has been shown elsewhere (e.g., DOI 10.1007/s11368-016-1512-4) that acidification of samples just desorbs the elements from particles, while elements that are associated with the matrix constituents (e.g., Fe, Al, Ti) cannot be decomposed in this way. Due to this reason, microwave-assisted digestion was suggested to be applied instead of acidification, in particular for the samples with high content of suspended particulate matter. Although the content of suspended particulate matter was relatively low for water samples from the Ob River (with average value of 21 mg/L), the sum of the suspended and dissolved concentrations (presented in Table 3) was for certain elements significantly lower than the “total” element concentrations in unfiltered and acidified water samples (presented in Table 2). This might indicate that not all elements were completely released from the suspended particles into the solution during acidification and that sample digestion might be necessary.
- I noticed some spelling errors or ambiguities in meaning (e.g. page 2: lines 74-77; page 3: line 113; page 5: lines 199-200, etc.) that need to be revised and corrected accordingly.
Author Response
Dear Rewiewer! Thank you for your detailed review of the manuscript. We have made corrections according to your comments.
All corrections are highlighted in green in the manuscript.
Responding in detail to the comments
Reviewer’s remarks |
Corrections |
Page 4: lines 167-175: There is some discrepancy in the reporting of certain elements that were determined by the two methods (i.e., ICP-OES or ICP-MS). Namely, the list of elements that were determined either by ICP-OES or ICP-MS method (reported in lines 167-173) do not completely match with the elements determined by both methods (listed in lines 174-175 |
Corrected |
- ICP-MS/AES instrumental parameters that were used for determination of element concentrations in river samples are missing and should be added to e.g. supplementary material.
|
See table SM2 for measurement parameters:
The MS measurements (ICP-MS (Х-7, Thermo Scientific, USA) were made using the standard parameters: a RF generator power of 1250 W; a PolyCon nebulizer; a plasma-forming Ar flow rate of 12 L/min; an auxiliary Ar flow rate of 0.9 L/min; an Ar flow rate into the nebulizer of 0.9 L/min; an analysed sample flow rate of 0.8 mL/ min
The AES measurements (iCAP-6500, Thermo Scientific, USA) were made using the standard parameters: a RF generator power of 1200 W; a PolyCon nebulizer; a plasma-forming Ar flow rate of 13 L/min; an auxiliary Ar flow rate of 0.8 L/min; an Ar flow rate into the nebulizer of 0.8 L/min; an analysed sample flow rate of 1.5 mL/ min
|
Page 6, lines 252-255: Have the authors considered that ICP-MS/AES analysis of river water samples that were just acidified might not reflect the total concentration of elements in the samples? It has been shown elsewhere (e.g., DOI 10.1007/s11368-016-1512-4) that acidification of samples just desorbs the elements from particles, while elements that are associated with the matrix constituents (e.g., Fe, Al, Ti) cannot be decomposed in this way. Due to this reason, microwave-assisted digestion was suggested to be applied instead of acidification, in particular for the samples with high content of suspended particulate matter. Although the content of suspended particulate matter was relatively low for water samples from the Ob River (with average value of 21 mg/L), the sum of the suspended and dissolved concentrations (presented in Table 3) was for certain elements significantly lower than the “total” element concentrations in unfiltered and acidified water samples (presented in Table 2). This might indicate that not all elements were completely released from the suspended particles into the solution during acidification and that sample digestion might be necessary. |
We agree with the reviewer's comment. The content of some elements in unfiltered samples was significantly (5-19 times) lower than the sum of suspended and dissolved concentrations. The content of sparingly soluble elements (Al, Zr, Ti, W) differed the most. Therefore, we excluded these elements from Table 2 and further analyses in section 3.2.1. For the remaining elements we obtained high or satisfactory agreement. For example, practically coincided content of elements in unfiltered samples with sum of suspended and dissolved concentrations of Ca, K, Mg, Na, Si, As, Ba, Cr, Mo, Co, Li
|
I noticed some spelling errors or ambiguities in meaning (e.g. page 2: lines 74-77; page 3: line 113; page 5: lines 199-200, etc.) that need to be revised and corrected accordingly |
Corrected Data on the content of trace elements in large river basins are necessary to assess the impact of climate change and land use; however, sufficiently large-scale studies of such a plan are rare [25]. page 5: lines 199-200, ИсправленоTo check the accuracy of measurements, we used multi-element certified reference materials Trapp ST-2a (Russian State Standard GSO 8671-2005).
|
Round 2
Reviewer 1 Report
Given the revisions made by the authors, I think the manuscript is now publishable.
Nevertheless, I would still emphasize that wet acid digestion performed in open vessels can be questionable anyway.